# DNA Methylation of α-Synuclein Intron 1 Is Significantly Decreased in the Frontal Cortex of Parkinson’s Individuals with *GBA1* Mutations

**DOI:** 10.3390/ijms24032687

**Published:** 2023-01-31

**Authors:** Adam R. Smith, David M. Richards, Katie Lunnon, Anthony H. V. Schapira, Anna Migdalska-Richards

**Affiliations:** 1Faculty of Health and Life Sciences, University of Exeter, Exeter EX2 5DW, UK; 2Living Systems Institute, University of Exeter, Exeter EX4 4QD, UK; 3Department of Physics and Astronomy, University of Exeter, Exeter EX4 4QD, UK; 4Department of Clinical and Movement Neurosciences, University College London Queen Square Institute of Neurology, London NW3 2PF, UK

**Keywords:** Parkinson’s disease, epigenetics, DNA methylation, DNA hypomethylation, α-synuclein (*SNCA*), glucocerebrosidase 1 (*GBA1*), GCase

## Abstract

Parkinson’s disease (PD) is a common movement disorder, estimated to affect 4% of individuals by the age of 80. Mutations in the glucocerebrosidase 1 (*GBA1*) gene represent the most common genetic risk factor for PD, with at least 7–10% of non-Ashkenazi PD individuals carrying a *GBA1* mutation (PD-*GBA1*). Although similar to idiopathic PD, the clinical presentation of PD-*GBA1* includes a slightly younger age of onset, a higher incidence of neuropsychiatric symptoms, and a tendency to earlier, more prevalent and more significant cognitive impairment. The pathophysiological mechanisms underlying PD-*GBA1* are incompletely understood, but, as in idiopathic PD, α-synuclein accumulation is thought to play a key role. It has been hypothesized that this overexpression of α-synuclein is caused by epigenetic modifications. In this paper, we analyze DNA methylation levels at 17 CpG sites located within intron 1 and the promoter of the α-synuclein (*SNCA*) gene in three different brain regions (frontal cortex, putamen and substantia nigra) in idiopathic PD, PD-*GBA1* and elderly non-PD controls. In all three brain regions we find a tendency towards a decrease in DNA methylation within an eight CpG region of intron 1 in both idiopathic PD and PD-*GBA1*. The trend towards a reduction in DNA methylation was more pronounced in PD-*GBA1*, with a significant decrease in the frontal cortex. This suggests that PD-*GBA1* and idiopathic PD have distinct epigenetic profiles, and highlights the importance of separating idiopathic PD and PD-*GBA1* cases. This work also provides initial evidence that different genetic subtypes might exist within PD, each characterized by its own pathological mechanism. This may have important implications for how PD is diagnosed and treated.

## 1. Introduction

Parkinson’s disease (PD) is the second most common neurodegenerative disorder after Alzheimer’s disease (AD). It is estimated to affect 4% of individuals by the age of 80 [1,2,3]. There is still remarkably little known about the causes of PD, and there are currently no treatments that can cure or modify the course of the disease. The treatments currently available are limited to improving symptoms [4]. In addition, at present, there are no conclusive lab-based tests for either diagnosis or prognosis, and diagnosis is essentially clinical [5].

Mutations of the glucocerebrosidase 1 (*GBA1*) gene, which encodes a lysosomal enzyme (GCase) that catabolizes glycolipid glucocerebroside to ceramide and glucose [6], comprise the most commonly known genetic cause of nonfamilial PD [7]. It has been estimated that at least 7–10% of non-Ashkenazi PD individuals have a *GBA1* mutation (denoted PD-*GBA1* individuals), with the most common mutations being L444P and N370S [3,7]. Interestingly, *GBA1* shows incomplete penetrance, with only about 30% of *GBA1*-mutation carriers developing the disease [3].

The clinical manifestation of PD-*GBA1* is similar to that of idiopathic PD, except for a slightly younger age of onset, a higher incidence of neuropsychiatric features (including sleep disturbance, hallucinations, depression and anxiety) and a greater risk for earlier and more prevalent cognitive impairment [7,8,9]. Notably, the pathology of PD-*GBA1* is identical to idiopathic PD, with the loss of dopaminergic neurons in the substantia nigra and the presence of Lewy bodies and neurites containing α-synuclein [8,10].

Although the exact molecular mechanisms by which *GBA1* mutations increase the risk of PD are still unknown, it is likely that, as in idiopathic PD, α-synuclein accumulation plays a central role in the development and progression of the disease [3]. This key role of α-synuclein in the pathology of PD-*GBA1* has been further highlighted by the reciprocal relationship between GCase activity and α-synuclein. Several studies have demonstrated that reduced GCase activity results in increased α-synuclein levels in SH-SY5Y cell cultures, neuronal cultures, conduritol-β-epoxide (CBE)-treated mice and transgenic *Gba1* mouse models [11,12,13,14,15,16,17,18,19,20]. Conversely, it has also been shown in cell and mouse models that increased α-synuclein causes a decrease in GCase activity [21,22].

Over the past decade, several putative mechanisms of α-synuclein accumulation in individuals both with idiopathic PD and PD-*GBA1* have been proposed, including lysosomal and mitochondrial dysfunction, endoplasmic reticulum stress, and impaired lipid metabolism, protein trafficking and α-synuclein clearance [3,23,24]. More recently, based on work in a heterozygous *Gba1* mouse model carrying a L444P knock-in mutation, it has been suggested that epigenetic modifications might cause α-synuclein overexpression, resulting in subsequent α-synuclein accumulation, which in turn could lead to significant loss of dopaminergic neurons in the substantia nigra [18]. Further, the putative role of epigenetics in the development of PD is supported by the low concordance rate for the disease between monozygotic twins, which is only about 17% [25].

Dynamic regulatory epigenetic processes, which control gene expression via modifications to DNA, histone proteins and chromatin, are essential for normal development and cell differentiation [26]. Dysregulation of epigenetic patterns has been widely regarded as relevant to the etiology of many neurodegenerative and neuropsychiatric disorders, including AD, Huntington’s disease (HD) and schizophrenia [26,27,28,29,30].

DNA methylation, the best characterized and most frequently studied epigenetic modification, influences gene expression by recruiting methyl-CpG-binding proteins and disrupting the binding of transcription factors, typically leading to gene silencing and chromatin remodeling [31]. DNA methylation plays an essential role in the brain, evidenced by the rapid changes in expression of the de novo DNA methyltransferases DNMT3A and DNMT3B during prenatal brain development [32] and by the occurrence of severe neurodevelopmental deficits as a consequence of mutations in the methyl-CpG binding protein 2 (*MECP2*) gene (which interacts with methylated DNA to control neuronal gene expression) [33]. Further, DNA methylation has been shown to play an important role in a number of neurodegenerative disorders such as AD, amyotrophic lateral sclerosis and HD [34,35].

The majority of studies of DNA methylation in PD brains conducted to date have focused on the core regulatory regions of the α-synuclein (*SNCA*) gene because of its clear implication in PD etiology. The results of these studies have been inconclusive, with some indicating no differences in DNA methylation levels, and others showing decreased or increased DNA methylation [36,37,38,39,40,41,42]. However, a major limitation of those studies is that they did not take into account genetic variation. Only Pihlstrøm et al. have attempted to investigate the interaction between genetics and DNA methylation [40], and this study neglected the role of *GBA1* gene.

In this study, for the first time, we measured DNA methylation in intron 1 and the promoter of the *SNCA* gene in the frontal cortex, putamen and substantia nigra of individuals with PD-*GBA1*. We also evaluated DNA methylation at the same sites and brain regions in individuals with idiopathic PD. We observed a significant decrease in DNA methylation within an eight CpG region of intron 1 in the frontal cortex of individuals with PD-*GBA1*, but not of individuals with idiopathic PD. These results suggest that PD-*GBA1* and idiopathic PD might have distinct epigenetic profiles. It also suggests that failure to separate PD-*GBA1* from idiopathic PD samples may explain why most previous studies have failed to find significant DNA methylation differences [36,37,38,39,40,41,42].

## 2. Results

To investigate whether DNA methylation differences might be present in idiopathic PD and PD-*GBA1*, we analyzed DNA methylation levels of 17 CpGs located within intron 1, as well as the promoter of *SNCA* in three different brain regions (the frontal cortex, putamen and substantia nigra).

### 2.1. Significant DNA Hypomethylation of Five CpG Sites of SNCA Intron 1 Located Further from the Transcription Start Site (TSS) in the Frontal Cortex of PD-GBA1

DNA methylation at all eight consecutive CpG sites of intron 1 (CpG1-8) located further from the TSS of *SNCA* showed a tendency towards a decrease in both idiopathic PD and PD-*GBA1* compared with elderly non-PD controls (Figure 1). DNA methylation was consistently lower at all eight sites in PD-*GBA1* compared with idiopathic PD, displaying significance at two CpG sites (CpG3, *p* = 0.0158 and CpG6, *p* = 0.0498) (Figure 1). More importantly, significant DNA hypomethylation was observed in five CpG sites (CpG2, *p* = 0.0406; CpG3, *p* = 0.0159; CpG5, *p* = 0.0321; CpG6, *p* = 0.0390 and Cpg8, *p* = 0.0473) in PD-*GBA1* samples compared to non-PD controls (Figure 1). No significant DNA methylation differences were observed between idiopathic PD and control samples (Figure 1). Altogether, this suggests that DNA hypomethylation at *SNCA* intron 1 in the frontal cortex might play a specific role in PD-*GBA1* and indicates the necessity of analyzing idiopathic PD separately from PD-*GBA1* samples. We also grouped the idiopathic PD and PD-*GBA1* samples together and analyzed the same eight sites against elderly non-PD controls. This is then directly comparable to all previous papers, none of which considered the genetic background [36,37,38,39,40,41,42]. No significant differences were observed, agreeing with much of the previous work, and further highlighting the importance of separating idiopathic PD from PD-*GBA1*. Finally, we compared the mean DNA methylation across all eight CpG sites for all the pair-wise combinations described above, but did not observe any significant differences.

We also assessed DNA methylation at a further six consecutive CpG sites of intron 1 (CpG9-14) located close to the TSS (Appendix A). We observed a trend towards an increase in DNA methylation at all sites in idiopathic PD and four CpG sites in PD-*GBA1* compared with elderly non-PD controls. No significant DNA methylation differences were observed between, one the one hand, elderly non-PD controls and, on the other, (a) idiopathic PD cases, (b) PD-*GBA1* cases, or (c) combined idiopathic PD and PD-*GBA1* cases. Furthermore, no significant DNA methylation differences were observed at any of the six CpGs between PD-*GBA1* and idiopathic PD. We also compared the mean DNA methylation across all six CpG sites between the same pairs of samples, but did not observe any significant differences.

### 2.2. A Trend towards a Decrease in DNA Methylation of CpG Sites in SNCA Intron 1 Further from the TSS in the Putamen of Idiopathic PD and PD-GBA1

We then analyzed the same 14 sites in the putamen (Appendix A). At sites CpG1-8 we observed a trend towards a decrease in DNA methylation at all eight sites in idiopathic PD and at five sites in PD-*GBA1* compared with elderly non-PD controls (Appendix A). However, no significant DNA methylation differences were observed between elderly non-PD controls and (a) idiopathic PD, (b) PD-*GBA1*, and (c) combined idiopathic PD and PD-*GBA1*. Also, no significant DNA methylation differences were observed at any of the eight CpGs between PD-*GBA1* and idiopathic PD. Finally, we compared the mean DNA methylation across all eight CpG sites between controls and (a) idiopathic PD, (b) PD-*GBA1*, (c) combined idiopathic PD and PD-*GBA1*, but again we did not observe any significant differences.

Next, we repeated this analysis of the putamen at the six CpG sites located close to the TSS (CpG9-14) but did not observe significant differences between any pairs of samples, including when idiopathic PD and PD-*GBA1* were combined and when all six sites were averaged over (Appendix A).

### 2.3. A Trend towards a Decrease in the DNA Methylation of CpG Sites in SNCA Intron 1 Further from the TSS in the Substantia Nigra of Idiopathic PD

In the substantia nigra at sites CpG1-8 we found that, compared to controls, methylation shows a tendency towards a decrease at all eight sites in idiopathic PD and increased at six sites in PD-*GBA1* (Appendix A). However, there were no significant differences for any pairs of samples. This did not change if all PD samples were combined together or for the average methylation across all eight sites. For the further six sites located close to the TSS (CpG9-14), there were also no significant differences (Appendix A).

### 2.4. No Significant DNA Methylation Differences within the SNCA Promoter in Either the Frontal Cortex or the Putamen

We then considered the DNA methylation of three CpG sites (Cpg15-17) within the promoter, which, in many genes, plays an important role in transcriptional regulation. We first examined the frontal cortex and putamen (Appendix A), but did not find any significant differences between (a) controls and idiopathic PD, (b) controls and PD-*GBA1*, or (c) idiopathic PD and PD-*GBA1*. Combining all PD samples and averaging over the three sites also did not show any significant differences associated with PD. These results (at least in the frontal cortex and putamen) suggest that it is the CpG sites closest to the TSS (i.e., CpGs within intron 1) that show the greatest methylation differences in Parkinson’s disease.

### 2.5. Significantly Lower DNA Methylation of CpG Site in the SNCA Promoter in the Substantia Nigra in Idiopathic PD but Not in PD-GBA1

Finally, we examined the same three promoter sites in the substantia nigra. Compared to controls, CpG16-17 showed a tendency towards a slight decrease in both PD and PD-*GBA1*, whilst the last site (CpG17) showed a trend towards a slight increase in methylation (Figure 2). However, the only significant results were at CpG16, where idiopathic PD was hypomethylated compared to both controls and PD-*GBA1* (*p* = 0.0015 and *p* = 0.0159, respectively). When we combined all PD samples together and compared to controls, we saw no significant differences when averaged over the amplicon.

Altogether, this points to the possibility that *SNCA* hypomethylation in the promoter might play a role in idiopathic PD, further highlighting the importance of analyzing idiopathic PD separately from PD-*GBA1*. However, it should be noted that this is based only on three CpG promoter sites and three non-PD controls. Future analysis including more CpG sites and more controls is necessary before firm conclusions can be drawn.

## 3. Discussion

This study provides the first measurement of DNA methylation of α-synuclein in individuals with Parkinson’s disease who carry *GBA1* mutations. We considered three brain regions (the frontal cortex, putamen and substantia nigra) and also assessed DNA methylation levels in the corresponding brain regions of individuals with idiopathic PD. For the first time, this allowed us to determine whether *SNCA* DNA methylation shows a unified pattern across the brain in PD.

We observed a tendency towards a decrease in DNA methylation in PD-*GBA1* and idiopathic PD samples within the region of *SNCA* intron 1 further from the TSS (eight consecutive CpG sites; CpG1-8) in the frontal cortex. However, significant DNA hypomethylation was only detected at five sites in individuals with PD-*GBA1* and at no sites in individuals with idiopathic PD. Combined analysis of all PD samples (PD-*GBA1* and idiopathic PD) also did not reveal any significant DNA methylation differences. This agrees with the results of Jowaed et al., who explored DNA methylation at the same eight consecutive CpGs in the frontal cortex of idiopathic PD and detected lower, but not significant, differences in DNA methylation levels at all eight sites [36]. In contrast, both Matsumoto et al., who analyzed the same eight CpGs in the anterior cingulate and de Boni et al., who analyzed six out of these eight CpGs in the cingulate gyrus and temporal frontal cortex, reported some of these CpGs to show a trend towards a decrease in DNA methylation and some to display a trend towards an increase in DNA methylation in idiopathic PD [37,38]. 

The fact that we found significant hypomethylation in PD-*GBA1* but not in idiopathic PD suggests that distinct epigenetic profiles may be present in these conditions. This would also explain why previous work, which has only ever considered mixed samples (of PD-*GBA1* and idiopathic PD), have often not found any significant methylation differences. If this is the case, this would strongly suggest the importance of separating idiopathic PD from PD-*GBA1* in future work. It is tempting to speculate that the need to segregate PD samples by genetic background may also extend to other PD-related genes (such as *LRRK2*, *PRKN* or *PINK1*), and it is possible that the true role of DNA methylation in PD will only be revealed once genetic and epigenetic analyses are considered in combination as a matter of routine.

Analysis of the same eight consecutive CpG sites in the putamen and the substantia nigra revealed a slight tendency towards a decrease in DNA methylation at all eight sites in idiopathic PD. In PD-*GBA1*, DNA methylation showed a trend towards a decrease at five CpG sites in the putamen, but displayed a slight tendency towards an increase at six sites in the substantia nigra. In the putamen, this agrees with Jowaed et al., although we did not replicate the significant decrease they reported at CpG7 [36], and with Matsumoto et al. and de Boni et al. [37,38]. In the substantia nigra, our results also match with those in de Boni et al. and Guhathakurta et al. [38,41], but we did not find the average methylation decrease described in Jowaed et al. and Matsumoto et al. [36,37].

Interestingly, we only observed significant DNA hypomethylation in the frontal cortex, with no significant differences found in the substantia nigra and putamen. Potentially, this could be due to differences in the pathological manifestation between those regions. Although the frontal cortex shows widespread Lewy body pathology in advanced PD, no prominent neurodegeneration is observed [43]. This should be contrasted to the substantia nigra and putamen where both Lewy body accumulation and substantial loss of nigrostriatal neurons occur [44]. Since the DNA methylation of a given region is a function of both the methylation profile of individual cell types and the ratio of cell types, it is plausible that (assuming DNA methylation differences occur mostly in neurons) selective loss of neurons might mask the true DNA methylation differences associated with PD. In short, it may be that neurons in the substantia nigra and putamen show the same hypomethylation as in the frontal cortex, but that neuronal loss in these regions hides this effect. This issue highlights the problem of profiling whole brain regions that consist of multiple cell types. To rectify this, it will be crucial in future work to separate nuclei of individual cell types before measuring DNA methylation.

Next, we examined a further six consecutive CpG sites (CpG9-14) within *SNCA* intron 1, situated close to the TSS. We did not observe any significant differences in PD-*GBA1*, idiopathic PD or combined samples in any of the analyzed regions. Our results are in agreement with previous studies conducted by Jowaed et al., de Boni et al. and Guhathakurta et al., who also found no significant differences [36,38,41]. However, it is worth noting that Jowaed et al. did observe a trend towards a decrease in DNA methylation at all six CpGs in putamen and five CpGs in the frontal cortex and substantia nigra [36]. We also analyzed three CpG sites (CpG15-17) located within the *SNCA* promoter. Again, we did not observe any significant differences in PD-*GBA1*, idiopathic PD or combined samples in either the frontal cortex or putamen. However, we did find a significant decrease at CpG 16 in the substantia nigra in idiopathic PD but not in PD-*GBA1*. It is worth noting that the only other paper to examine the substantia nigra within the promoter analyzed different CpG sites [42]. These data further highlight the importance of separating idiopathic PD from PD-*GBA1*: combined analysis is likely to miss important methylation differences. Although based on only a small sample size, our results could be an early indication that CpG sites within the promoter are important in idiopathic PD, whereas sites within intron 1 are important in PD-*GBA1*.

As in previous studies, we also examined the average DNA methylation across all analyzed CpG sites within *SNCA* intron 1 and all sites within the promoter. However, we did not observe any significant differences. This agrees with Matsumoto et al. and Guhathakurta et al. [37,41] but not with Jowaed at al. (who found significant hypomethylation in intron 1 in the frontal cortex, putamen and substantia nigra) or with de Boni et al. (who found significant hypermethylation in intron 1 in the putamen, although only in Braak stages 3 and 4) [36,38]. However, it is important to note that Jowaed at al. and de Boni et al. compared average DNA methylation across 23 and 19 CpG sites, respectively, whereas we only considered 14 CpG sites.

One limitation of our study (along with all previous studies) is the relatively small sample size. It will be important to address this in the future. This might uncover not only the presence of more significant DNA hypomethylation in PD-*GBA1* samples, but also allow potential discrimination between idiopathic PD and PD-*GBA1* samples. Further, it would also be interesting to investigate the association of DNA methylation with clinical disease duration. In our study we analyzed the frontal cortex, putamen and substantia nigra, but in future it may be important to include other brain regions involved in PD (such as the thalamus and amygdala) as well as to examine smaller brain subregions (such as the anterior, frontal and temporal cortices).

Finally, as it has been shown that only about 30% of individuals with *GBA1* mutations develop PD by the age of 80 [3], it will in future also be important to compare the DNA methylation profile between PD-*GBA1* and non-PD-*GBA1* samples. Such an approach might potentially reveal novel epigenetic mechanisms that orchestrate the transition from a healthy *GBA1*-mutation carrier to an individual with PD. This is supported by recent work in a heterozygous *GBA1* mice carrying a L444P knock-in mutation, where it was proposed that epigenetic modifications might lead to α-synuclein overexpression and its subsequent accumulation, which in turn could result in significant loss of nigral dopaminergic neurons [18].

## 4. Materials and Methods

### 4.1. Subjects and Samples

Post-mortem brain tissue samples were obtained from the Queen Square Brain Bank for Neurological Disorders, which provided ethical approval for the use of the samples (Ref. No. 18/LO/0721). Tissues were obtained from the frontal cortex, putamen and substantia nigra as these are the regions of most prominent pathology in PD [44]. For the frontal cortex, samples were obtained from 6 elderly non-PD control subjects (free of any neurological conditions), 11 idiopathic PD subjects and 9 PD-*GBA1* subjects. For the putamen, samples were obtained from 6 elderly non-PD control subjects (free of any neurological conditions), 9 idiopathic PD subjects and 6 PD-*GBA1* subjects. For the substantia nigra, samples were obtained from 3 elderly non-PD control subjects (free of any neurological conditions), 13 idiopathic PD subjects and 8 PD-*GBA1* subjects. Genomic DNA was isolated from 100 mg of each dissected brain region using a standard phenol-chloroform extraction method and tested for degradation and purity before analysis as described previously [45]. Demographic information for all samples can be found in Table 1.

### 4.2. SNCA Bisulfite Pyrosequencing

Bisulfite pyrosequencing was used to quantify DNA methylation across 17 individual CpG sites in the promoter and intron 1 of the *SNCA* gene, which is located within region 90,757,151 to 90,758,651 within chromosome 4 (hg19) (Figure 3). Bisulfite conversion was performed with the Bisulfite-Gold kit (Zymo research, USA). Three separate amplicons were generated using primers designed using the PyroMark Assay Design software 2.0 (Qiagen, UK) (Table 2). Pyrosequencing was performed with the pyrosequencing primers listed in Table 2. DNA methylation was quantified using the Pyromark Q24 system (Qiagen, UK), following the manufacturer’s standard instructions, and the Pyro Q24 CpG 2.0.6 software, providing a percentage methylation score for each CpG site.

### 4.3. Data Analysis

All computational and statistical analyses were performed using a custom-built R 3.3.2 script (R Development Core Team, 2012). A linear model analysis examined DNA methylation differences at individual CpG sites, and was performed controlling for the effects of age and sex, comparing (a) control samples with PD-*GBA1* samples, (b) control samples with idiopathic PD samples, (c) controls samples with combined PD-*GBA1* and idiopathic PD samples, and (d) PD-*GBA1* samples and idiopathic PD samples.

## 5. Conclusions

We showed that DNA methylation in the frontal cortex shows a tendency towards a decrease in both PD-*GBA1* and idiopathic PD within part of *SNCA* intron 1. Interestingly, we found a greater decrease in PD-*GBA1* than in idiopathic PD samples. For the first time, we revealed that DNA methylation in the frontal cortex is significantly lower within part of intron 1 in individuals with PD-*GBA1*, but not in individuals with idiopathic PD. These results suggest that a potentially distinct epigenetic profile is present in PD-*GBA1* compared to idiopathic PD, which in turn might lead to increased accumulation of α-synuclein in PD-*GBA1*.

Our results provide a tentative explanation for why previous work has found only a handful of significant DNA methylation differences in PD: analyses that mix PD-*GBA1* and idiopathic PD samples might dilute any observed differences. Therefore, based on our work here, we strongly advocate the importance in future of not only separating idiopathic PD and PD-*GBA1* samples, but extending this to the stratification of PD samples by other known PD genes. We believe only then will we be able to uncover the true DNA methylation differences involved in PD.

Finally, our work provides an initial indication that, contrary to what has been previously assumed, PD might actually be a cluster of distinct genetic subtypes, each characterized by its own pathophysiological mechanism. Such a finding might not only fundamentally alter our understanding of PD, but, more importantly, would have substantial implications for the development of future diagnostic biomarkers and treatments.

## Figures and Tables

**Figure 1 ijms-24-02687-f001:**
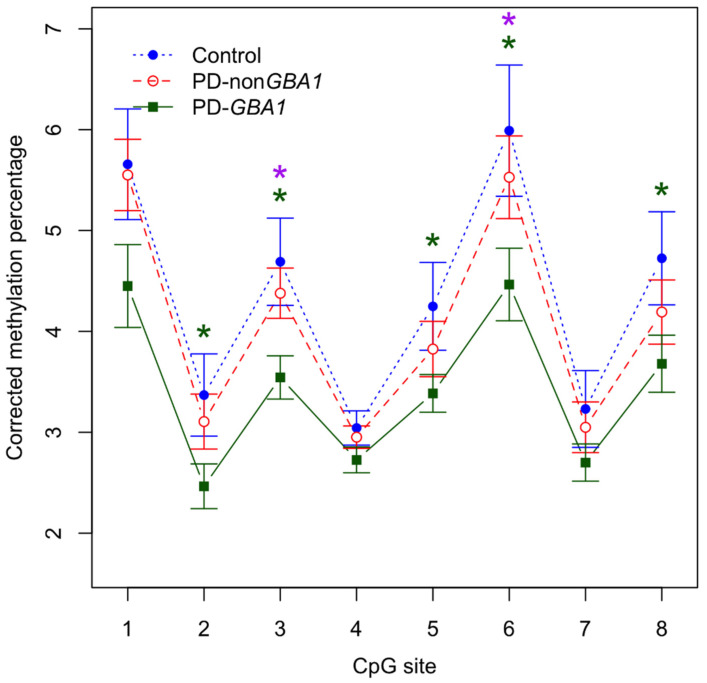
Significant DNA hypomethylation in the frontal cortex in PD-*GBA1*. The methylation profile within *SNCA* intron 1 in the frontal cortex shows a significant decrease (in 5 out of 8 analyzed CpG sites; green stars) for individuals with PD-*GBA1* (green) compared to matched controls (blue). DNA methylation was also significantly lower (in 2 out of 8 analyzed CpG sites; purple stars) for individuals with PD-*GBA1* (green) compared to idiopathic PD cases (red). *n* = 11 for PD-non*GBA1*, *n* = 10 for PD-*GBA1*, *n* = 7 for controls. Legend: * = *p* < 0.05, PD-non*GBA1* = idiopathic PD. Error bars–SD. Corrected methylation percentage adjusted for age and gender.

**Figure 2 ijms-24-02687-f002:**
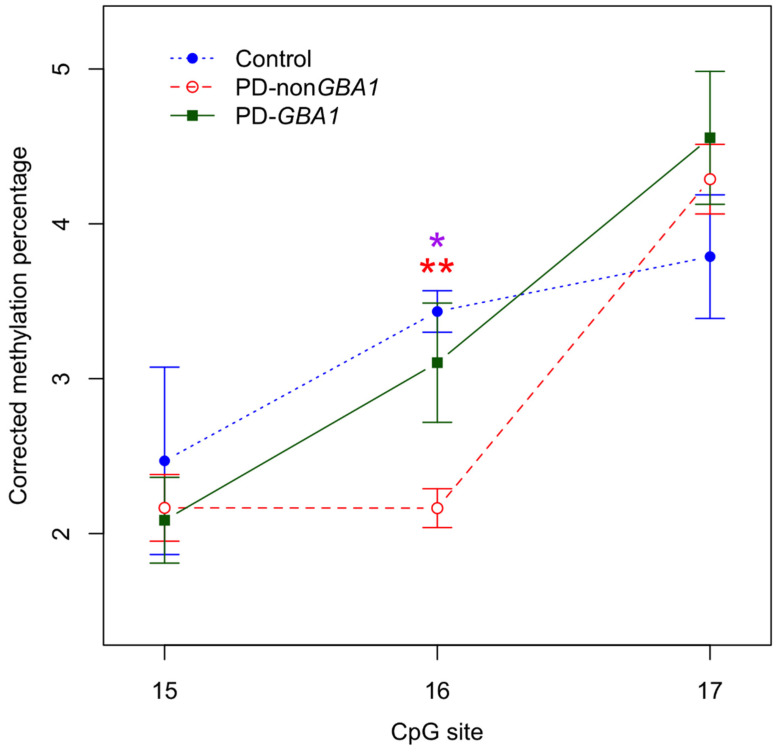
Significant DNA hypomethylation within the *SNCA* promoter in the substantia nigra in idiopathic PD. The methylation profile within the *SNCA* promoter in the substantia nigra shows significant decrease at CpG16 (red stars) for individuals with PD-non*GBA1* (red) compared to matched controls (blue). DNA methylation was also significantly lower at the same site (purple star) for individuals with PD-non*GBA1* (red) compared to PD-*GBA1* cases (green). *n* = 13 for PD-non*GBA1*, *n* = 6 for PD-*GBA1*, *n* = 3 for controls. Legend: ** = *p* < 0.01, * = *p* < 0.05, PD-non*GBA1* = idiopathic PD. Error bars–SD. Corrected methylation percentage adjusted for age and gender.

**Figure 3 ijms-24-02687-f003:**
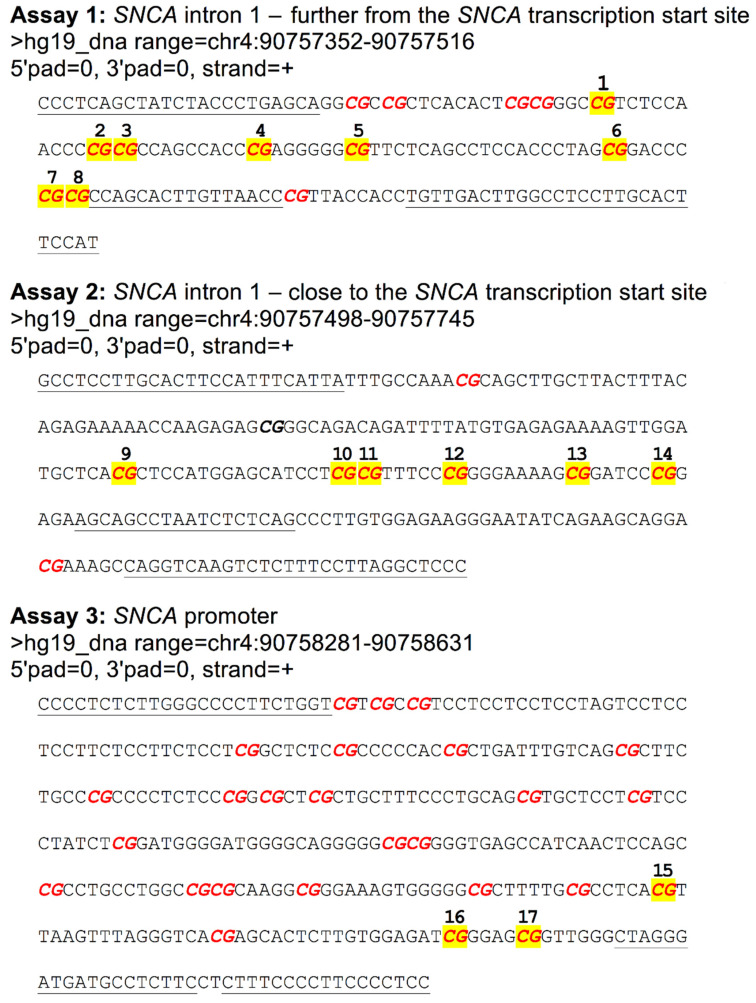
Location of the measured CpG sites within the promoter and intron 1 of the *SNCA* gene (which is encoded on the minus strand of chromosome 4). Legend: In bold red—CpG sites; yellow highlight—CpG sites used in this work; underline—annealing sequences of PCR primers and pyrosequencing primers.

**Table 1 ijms-24-02687-t001:** Demographic information for all analyzed samples. Legend: PD–Parkinson’s disease, PMD–post-mortem delay, FC–frontal cortex, PUT–putamen, SN–substantia nigra, X–sample pyrosequenced in this work.

Sex	Age	PMD	Cause of Death	FC	PUT	SN	*GBA1* Mutation
**Control cases**							
F	81	44.55	Carcinoma (breast)	X			-
M	95	89.20	Heart failure	X	X		-
M	87	39.25	Bronchopneumonia	X	X	X	-
M	63	42.0	Congestive heart disease	X		X	-
F	64	79.0	Carcinoma (bowel)		X		-
F	53	29.5	Intra-cerebral hemorrhage		X		-
F	82	77.25	Unknown	X	X	X	-
M	43	15.00	Heart failure	X			-
M	71	38.50	Mesothelioma	X			-
M	38	80.35	Carcinoma (gastric)		X		-
**Idiopathic PD cases**							
F	55	31.20	Idiopathic PD	X	X	X	-
M	67	35.10	Myocardial infarct			X	-
M	66	43.10	Pneumonia	X		X	-
F	61	25.30	End stage idiopathic PD		X		-
M	91	36.20	Idiopathic PD, dementia	X		X	-
M	85	28.30	PD, immobility	X		X	-
M	65	31.20	Aspiration pneumonia		X	X	-
M	68	38.50	Gradual deterioration	X	X	X	-
M	75	28.00	Chest infection	X	X	X	-
M	63	69.00	Heart failure	X	X	X	-
F	62	46.20	Gradual deterioration	X		X	-
M	59	94.00	Chest infection			X	-
M	66	30.00	Atypical PD, gradual deterioration	X	X	X	-
M	81	49.00	Aspiration pneumonia	X	X	X	-
M	65	55	Unknown	X	X		-
***PD-GBA1* cases**							
M	62	34.00	Progressive deterioration	X	X		G193E
M	55	8.00	Progressive degeneration	X			R131C
F	67	45.00	Myocardial infarct	X			L444P
M	57	28.00	Bronchopneumonia	X	X	X	RecA456P
F	64	Unknown	Unknown	X			
M	91	60.45	Ischemic bowel, sepsis			X	N370S
M	85	48.15	Chest infection				N370S
M	68	86.15	PD, dementia		X	X	R463C
M	62	46.00	Cardiac arrest	X		X	D409H
F	64	92.00	Respiratory failure	X	X	X	Rec*NciI*
M	59	42.45	Multiple system atrophy	X	X	X	L444P
F	57	85.40	Bronchopneumonia	X		X	L444P
M	82	54.40	Congestive heart failure		X	X	N370S

**Table 2 ijms-24-02687-t002:** Annealing sequences of PCR and pyrosequencing primers.

Assay	PCR Primers	Length	Pyrosequencing Primers
1	F: 5′-TGTTGACTTGGCCTCCTTGCACTTCCAT-3′ R: 5′-CCCTCAGCTATCTACCCTGAGCA-3′	164 bp	5′-CCAGCACTTGTTAACCC-3′
2	F: 5′-CAGGTCAAGTCTCTTTCCTTAGGCTCCC-3′ R: 5′-GCCTCCTTGCACTTCCATTTCATTA-3′	247 bp	5′-AGCAGCCTAATCTCTC-3′
3	F: 5′-CTTTCCCCTTCCCCTCCC-3′ R: 5′-CCCCTCTCTTGGGCCCCTTCTGGT-3′	350 bp	5′-CTAGGGATGATGCCTCTTC-3′

## Data Availability

The data supporting the results can be found in Appendix A.

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
