# Peer review of "DNA Methylation of α-Synuclein Intron 1 Is Significantly Decreased in the Frontal Cortex of Parkinson’s Individuals with GBA1 Mutations"

_ijms, 2023, doi:10.3390/ijms24032687_

Round 1

Reviewer 1 Report

The MS has been carefully analyzed. Please note my specific comments below:

Figure 1: Please add the unit in which the DNA methylation level has been calculated/estimated. Please delete the inter-CpG lines. Consider replacing the brown star with a different color to visualize the difference in a better fashion.

Figure 2: Please add the unit in which the DNA methylation level has been calculated/estimated. Please delete the inter-CpG lines. There is big difference between the sample size of the single groups ( from 13 to 3), please explain why. This reviewer would expect to the at least n=5, or better n=10 samples to generate a reliable statistical trend/value.

L336: The age range of the dead patients, from whom samples were generated is very large. This reviewer highly recommends and completely new analysis which respects age groups, eg. 50-65, 66-75, 76-85, 86-95), what especially would be of high importance for the epigentic changes. Besides, there should be the information provided, for how long each patient was suffering from PD.

Author Response

We thank the Reviewer for their careful analysis of the manuscript. Please find our response to specific comments below.

  1. “Figure 1: Please add the unit in which the DNA methylation level has been calculated/estimated. Please delete the inter-CpG lines. Consider replacing the brown star with a different color to visualize the difference in a better fashion.”

We have adjusted all the figures in the manuscript to better represent the data presented by changing the Y axis wording to “Corrected methylation percentage”.

We have linked the points between individual CpGs, as we believe it makes it easier for the reader to see differences between groups across the amplicon. We have used this method of plotting in pyrosequencing plots in our previous publications.(1-3) Finally, we have changed the brown star denoting significance to purple to aid visualisation. 

  1. “Figure 2: Please add the unit in which the DNA methylation level has been calculated/estimated. Please delete the inter-CpG lines. There is big difference between the sample size of the single groups ( from 13 to 3), please explain why. This reviewer would expect to the at least n=5, or better n=10 samples to generate a reliable statistical trend/value.”

For answer to the first part of your comment, please see our answer to your comment 1. As per sample size, we appreciate this valuable comment and agree with the Reviewer that a variable sample size is a limitation of our study. However, given the limited nature of post-mortem brain material from PD individuals and appropriate control individuals, this is the extent of available tissue for this study. Nevertheless, we do acknowledge this as a limitation of the study in the discussion.

  1. “L336: The age range of the dead patients, from whom samples were generated is very large. This reviewer highly recommends and completely new analysis which respects age groups, eg. 50-65, 66-75, 76-85, 86-95), what especially would be of high importance for the epigentic changes. Besides, there should be the information provided, for how long each patient was suffering from PD.”

We agree with this limitation of our study, and it is something we are keen to address with future work, however these were the only available samples for this study. Splitting the cohort by age is an interesting suggestion, however given our low sample size this will diminish our statistical power even further. To control for this issue, we have used age as a covariate in our analyses, so are confident that age has not influenced our findings. Unfortunately, no further clinical or pathological information is available for the samples so we are unable to stratify by disease duration in our analyses. However, we do suggest this as a future direction in our discussion.

References

  1. Smith AR, et al. Parallel profiling of DNA methylation and hydroxymethylation highlights neuropathology-associated epigenetic variation in Alzheimer's disease. Clin Epigenetics. 2019 Mar 21;11(1):52. doi: 10.1186/s13148-019-0636-y.
  2. Smith AR, et al. A cross-brain regions study of ANK1 DNA methylation in different neurodegenerative diseases. Neurobiol Aging. 2019 Feb;74:70-76. doi: 10.1016/j.neurobiolaging.2018.09.024.
  3. Lunnon K, et al. Methylomic profiling implicates cortical deregulation of ANK1 in Alzheimer's disease. Nat Neurosci. 2014 Sep;17(9):1164-70. doi: 10.1038/nn.3782.

Reviewer 2 Report

The manuscript “DNA methylation of α-synuclein intron 1 is significantly decreased in Parkinson’s individuals with GBA1 mutations”  compares methylation of 17 CpGs within the 1st intron and promoter of SNCA in the frontal cortex, putamen and substantia nigra in post mortem material from idiopathic PD, PD-GBA1 and control patients. The Authors identify lower level of methylation of 5 CpGs within the 1st intron of SNCA in the cortex of people with mutations in GBA1 (PD-GBA1).  

Although I agree with the Authors as to the “need to segregate PD samples by genetic background” I have some critical  remarks regarding the methodology and data interpretation

Methodology:

1. There should be more information in the Methods on the analysis and quantification of CpG methylation. For example, what was the average number of reads per amplicon/sample?

2. How to interpret the description of the Y axes in the figures, i.e. what does it mean that the “corrected methylation level” was, for example, 5? Also, what is the rationale for linking the points corresponding to the mean values of methylation of individual CpGs?

3. CpGs 1-8 and 9-14 in are located in the same region of intron I and should be analyzed together in Fig.1 even if, technically, they were on two different amplicons.

4. Judging from the scheme in figure 3 the gene promoter region  of SNCA is CpG  rich. Why only 3 CpGs were analyzed for the promoter region when there are numerous CpGs in the same amplicon?

Data interpretation:

1. Authors tend to refer to an increase/decrease in CpG methylation in a given CpG site or gene region even when  differences between the studied groups are not statistically significant, which is simply misleading as this can be only called a tendency. E.g., line 20-21 in the Abstract, they claim that the methylation was decreased in 3 brain regions while this is true only for the frontal cortex. Also, the statement in lines 110-112 claiming that methylation “was decreased in both idiopathic PD and PD-GBA1” is contradictory to the sentence in lines 117-118 stating that “no significant DNA methylation changes were observed  between idiopathic PD and control samples”. There are many other erroneous descriptions of this kind in the text.

2. In most cases the Authors should not refer to changes in methylation (eg. line 154) but to differences in methylation between the studied patient groups or brain regions. Also, increased/decreased should be substituted by higher/lower since they do not study dynamic changes within the same gene/brain region or patient group but compare independent samples.

Author Response

We thank the Reviewer for their comments and feedback. Please find below a breakdown of how we have addressed these in the manuscript.  

Methodology:

  1. There should be more information in the Methods on the analysis and quantification of CpG methylation. For example, what was the average number of reads per amplicon/sample?”

We apologise for this oversight. We have now provided this information by adding an extra line to the methods section to explain the output of the Pyro Q24 CpG software. As this is pyrosequencing rather than next-gen (Illumina) sequencing calculation of read depth per amplicon/ sample is not possible or appropriate.  

  1. How to interpret the description of the Y axes in the figures, i.e. what does it mean that the “corrected methylation level” was, for example, 5? Also, what is the rationale for linking the points corresponding to the mean values of methylation of individual CpGs?”

We apologise for the confusion over the term corrected methylation level. We agree that the Y axis description is misleading, so we have updated all figures to include “Corrected methylation percentage”. We calculated the corrected methylation level as the intercept from an initial model that adjusted for age and gender. We have now clarified this in the methods. Linking of the points was performed to graphically represent the methylation levels across the region as CpG sites are shown in genomic location order. We believe it makes it easier for the reader to see differences between groups across the amplicon. We have used this method of plotting in pyrosequencing plots in our previous publications as this is considered standard practice for methylation data generated in this manner.(1-3). Please also see our response to Reviewer 1, comments 1 and 2 as that Reviewer raised the same point.

  1. “CpGs 1-8 and 9-14 in are located in the same region of intron I and should be analyzed together in Fig.1 even if, technically, they were on two different amplicons.

We appreciate this valuable suggestion. However, we respectfully disagree with the Reviewer’s comment, as the two amplicons were amplified using separate PCRs under different conditions, so it is not appropriate to analyse the data together as suggested.  

  1. “Judging from the scheme in figure 3 the gene promoter region of SNCA is CpG  rich. Why only 3 CpGs were analyzed for the promoter region when there are numerous CpGs in the same amplicon?”

Due to the need to amplify the bisulfite converted DNA via PCR before pyrosequencing, assays are reliant on finding suitable PCR primers that do not overlap with a variable region (SNP or CpG site). As a result of the large numbers of CpGs in this area, as noted by the Reviewer, we were unable to assess any further sites in the region. This is a key limitation of pyrosequencing that we will look to address in future work.  

Data interpretation:

  1. “Authors tend to refer to an increase/decrease in CpG methylation in a given CpG site or gene region even when differences between the studied groups are not statistically significant, which is simply misleading as this can be only called a tendency. E.g., line 20-21 in the Abstract, they claim that the methylation was decreased in 3 brain regions while this is true only for the frontal cortex. Also, the statement in lines 110-112 claiming that methylation “was decreased in both idiopathic PD and PD-GBA1” is contradictory to the sentence in lines 117-118 stating that “no significant DNA methylation changes were observed between idiopathic PD and control samples”. There are many other erroneous descriptions of this kind in the text.”

We apologise for this misleading terminology. We have now corrected the terminology throughout the manuscript, and clearly indicated where DNA methylation was significantly changed and where only a tendency towards a change was observed.

  1. “In most cases the Authors should not refer to changes in methylation (eg. line 154) but to differences in methylation between the studied patient groups or brain regions. Also, increased/decreased should be substituted by higher/lower since they do not study dynamic changes within the same gene/brain region or patient group but compare independent samples.”

Again, we apologise for this misleading terminology. We have now corrected the terminology throughout the manuscript by changing “changes” to “differences” and substituting “increased/decreased” with “higher/lower”.

References

  1. Smith AR, et al. Parallel profiling of DNA methylation and hydroxymethylation highlights neuropathology-associated epigenetic variation in Alzheimer's disease. Clin Epigenetics. 2019 Mar 21;11(1):52. doi: 10.1186/s13148-019-0636-y.
  2. Smith AR, et al. A cross-brain regions study of ANK1 DNA methylation in different neurodegenerative diseases. Neurobiol Aging. 2019 Feb;74:70-76. doi: 10.1016/j.neurobiolaging.2018.09.024.
  3. Lunnon K, et al. Methylomic profiling implicates cortical deregulation of ANK1 in Alzheimer's disease. Nat Neurosci. 2014 Sep;17(9):1164-70. doi: 10.1038/nn.3782.

Round 2

Reviewer 1 Report

No further comments.

Author Response

We thank the Reviewer for their work.

Reviewer 2 Report

The manuscript has been improved and is now more substantive and readable. Nonetheless there are still several issues to be considered.

1. I still see no point in dividing the intron into 2 regions and naming one ” close to the transcription start site” (n.b. the abbreviation TSS should be used) and the other “further from the “ if they are separated  by about 100 nucleotides which is not really an important distance in relation to TSS. Also, if methylation is calculated and presented in the same units for both “regions”(i.e. as percentage of methylation) there is no reason why the results cannot be presented on the same graph.

2. I noticed that the genomic positions of the SNCA promoter (fig.3)  are downstream from those of the intron. Since the gene is on the + strand they should be upstream. Are the Authors sure that they analyzed SNCA promoter sequence?

Minor

- I still think there is no rationale in linking individual points in the figures as it only obscures the picture; this is however a minor issue and the Authors may stick to their view.

-abbreviation TSS is introduced later (l. 136) than the full term is used (line 117).

-the subtitles are very long. TSS should be used or, better still,  the ” near/further” terms should be totally deleted (see point 1.). As it is, to a non-native speaker at least, they  read as if TSS was in the putamen. E.g.

“A trend towards a decrease in DNA methylation of CpG sites in SNCA intron 1 near the  transcription start site in the putamen of idiopathic PD and PD-GBA1” vs “A trend towards a decrease in DNA methylation of CpG sites in intron 1 of SNCA in the putamen of idiopathic PD and PD-GBA1”

Round 3

Reviewer 2 Report

I still have doubts as to whether the Authors correctly mapped the analyzed SNCA gene fragments (and therefore correctly interpret the results). Namely, in Fig.3, the genomic co-ordinates of  intron 1 fragment described as "further from the TSS" are closer to those of the promoter than those of the intron 1 fragment described  as "close to the TSS. Since the order 5'-3' is : gene promoter, TSS, (exon1), intron 1 (close to TSS), intron 1 (far from TSS), there must be a mistake. 

Author Response

We thank the Reviewer for their careful revision of the manuscript and for their further comments and feedback. Please find our response to your specific comment below. 

I still have doubts as to whether the Authors correctly mapped the analyzed SNCA gene fragments (and therefore correctly interpret the results). Namely, in Fig.3, the genomic co-ordinates of intron 1 fragment described as "further from the TSS" are closer to those of the promoter than those of the intron 1 fragment described  as "close to the TSS. Since the order 5'-3' is : gene promoter, TSS, (exon1), intron 1 (close to TSS), intron 1 (far from TSS), there must be a mistake.” 

After reviewing all assays, rather than assay 3 in isolation, we now understand the Reviewer’s concerns and have accordingly adjusted the description of these assays (Figure 3) and our interpretation of the results.